# MESMERIC: Machine Learning-Based Trust Management Mechanism for the Internet of Vehicles

**DOI:** 10.3390/s24030863

**Published:** 2024-01-29

**Authors:** Yingxun Wang, Adnan Mahmood, Mohamad Faizrizwan Mohd Sabri, Hushairi Zen, Lee Chin Kho

**Affiliations:** 1Faculty of Engineering, Universiti Malaysia Sarawak, Kota Samarahan 94300, Sarawak, Malaysia; msmfaizrizwan@unimas.my (M.F.M.S.); lckho@unimas.my (L.C.K.); 2Faculty of Computer and Information Engineering, Qilu Institute of Technology, Jinan 250200, China; 3School of Computing, Macquarie University, Sydney, NSW 2109, Australia; adnan.mahmood@mq.edu.au; 4Faculty of Engineering and Technology, i-CATS University College, Kuching 93350, Sarawak, Malaysia; hushairi@icats.edu.my

**Keywords:** Internet of Vehicles, machine learning, trust management mechanism, direct trust, indirect trust, context, optimal decision boundary

## Abstract

The emerging yet promising paradigm of the Internet of Vehicles (IoV) has recently gained considerable attention from researchers from academia and industry. As an indispensable constituent of the futuristic smart cities, the underlying essence of the IoV is to facilitate vehicles to exchange safety-critical information with the other vehicles in their neighborhood, vulnerable pedestrians, supporting infrastructure, and the backbone network via vehicle-to-everything communication in a bid to enhance the road safety by mitigating the unwarranted road accidents via ensuring safer navigation together with guaranteeing the intelligent traffic flows. This requires that the safety-critical messages exchanged within an IoV network and the vehicles that disseminate the same are highly reliable (i.e., trustworthy); otherwise, the entire IoV network could be jeopardized. A state-of-the-art trust-based mechanism is, therefore, highly imperative for identifying and removing malicious vehicles from an IoV network. Accordingly, in this paper, a machine learning-based trust management mechanism, MESMERIC, has been proposed that takes into account the notions of direct trust (encompassing the trust attributes of interaction success rate, similarity, familiarity, and reward and punishment), indirect trust (involving confidence of a particular trustor on the neighboring nodes of a trustee, and the direct trust between the said neighboring nodes and the trustee), and context (comprising vehicle types and operating scenarios) in order to not only ascertain the trust of vehicles in an IoV network but to segregate the trustworthy vehicles from the untrustworthy ones by means of an optimal decision boundary. A comprehensive evaluation of the envisaged trust management mechanism has been carried out which demonstrates that it outperforms other state-of-the-art trust management mechanisms.

## 1. Introduction

The rapid acceleration in urbanization and growth of the population has substantially increased the ownership of vehicles. A conservative estimate by the World Health Organisation suggests that traffic accidents kill approximately 1.35 million people every year and approximately 50 million people suffer from non-fatal injuries [1]. Furthermore, the congestion of traffic is a global issue which also results in increased noise pollution and vehicular emissions [2,3]. Accordingly, numerous researchers from academia and industry over the years have focused on resolving such issues. Ensuring both passenger road safety and traffic congestion mitigation, therefore, requires an intelligent system of vehicular communication.

Vehicles today are an indispensable constituent of the Internet of Things (IoT) network and are accordingly equipped with hundreds of sensors onboard [4]. As per an estimate, modern vehicles are equipped with approximately 100 sensors onboard with each vehicle capable of producing nearly 380 TB to 4.9 PB data annually [5]. Therefore, vehicle-mounted sensors (position, velocity, acceleration, pressure, and temperature sensors) and IoT devices would be able to construct a safe and efficient intelligent network of transportation [6].

The IoV is an application of IoT in the context of intelligent transportation systems (ITS). The IoV has a similar architecture to the IoT and features a hierarchical structure that includes data source, edge, fog, and the cloud layers [7]. Vehicles share information with other vehicles, pedestrians, intelligent infrastructure, and backbone networks to establish vehicle-to-vehicle, vehicle-to pedestrian, vehicle-to-infrastructure, and vehicle-to-network communication, thereby formulating vehicle-to-everything (V2X) communication. Figure 1 thus depicts the architecture of an IoV network. The main IoV communication node is a vehicle with an on-board unit (OBU) which can communicate with the Roadside Units (RSUs) and other vehicles in its proximity. Due to the unique characteristics of IoV, i.e., openness, dynamic topology, and high mobility, it is susceptible to attacks; dishonest entities can modify legitimate security messages, spread forged information, or delay forwarding messages, thereby endangering human lives [8].

Accordingly, researchers have proposed several solutions for handling the issues pertinent to IoV security. Nevertheless, a number of these solutions rely on conventional cryptographic-related schemes and, therefore, rely on the notions of digital signatures, certificates, and public key infrastructure [9,10]. Moreover, conventional cryptographic-related schemes are only capable of mitigating external attacks and are ineffective against internal network attacks [11]. It is due to this reason that the paradigm of trust has been recently introduced in the research literature.

The notion of trust originated in sociology as a means to understand how people are interdependent within a social organization [12]. Trust, over the years, has also been employed in various other disciplines, including, but not limited to, philosophy, economics, engineering, and computer science. Trust is generally referred to as the confidence of a trustor in a trustee. Here, trustor refers to a node that is in a position to ascertain the trust of the other node (trustee) in the network, whereas the trustee refers to a node whose trust is being ascertained [13]. In the context of this paper, trust refers to the likelihood that a trustee can perform a particular operation (contribute to realizing a particular application or service) within a specific situation at a specific time. It is also important to mention that trust computation primarily involves a weighted aggregation of both the direct trust and the indirect trust [14]. Direct trust is ascertained as a result of direct interactions between a trustor and a trustee and is generally referred to as a trustor’s direct observation of a trustee [15,16]. On the contrary, indirect trust is computed by taking into account the direct trust ascertained by the one-hop neighbors of a trustor pertinent to a trustee. The literature argues that direct trust is more significant in contrast to indirect trust [1].

To date, a number of trust management models have been proposed in the research literature which have been broadly classified into three types: entity-oriented trust models, data-oriented trust models, and hybrid trust models [17]. Entity-oriented trust models aim to eradicate malicious entities (vehicles) from an IoV network by evaluating the reliability of the vehicles disseminating messages. Data-oriented trust models, on the other hand, eradicate the malicious messages instead of the entities from an IoV network [18]. Finally, hybrid trust models take into account the salient characteristics of both the entity-oriented and data-oriented trust models and, therefore, regard the reliability of the entities and the respective messages disseminated by them in a bid to make a decision.

The trust parameters, also referred to as the trust attributes, are integral constituents of any trust model. The existing literature suggests that a number of research studies determined the global trust value of a particular vehicle in an IoV network by taking into account the weighted sum of a number of such trust parameters and which, in fact, is also subject to limitations [19]. For instance, weights’ settings are based on human subjectivity and, therefore, different researchers often set different weights for the same trust parameter which results in an inconsistent trust score. Keeping this in mind, the envisaged research employs the notion of machine learning to ascertain trustworthy and untrustworthy vehicles via an optimal trust boundary. In order to better express the running situation of vehicles and achieve the optimal trust evaluation results, the trust model needs more trust parameters, but it will lead to an increase in the amount of calculation, so we can use the learning method to train the trust model. In this way, the global trust value of each vehicle was established by combining all of its respective trust parameters such that the optimal influence of each trust parameter on the global trust value is prevalent.

Also, a number of trust models do not take into account the effect of context while ascertaining the trust of a particular vehicle. For instance, an urban scenario involves more vehicles and, therefore, the inter-vehicular interactions are much more as opposed to a highway scenario, wherein it is extremely challenging to establish trust among the vehicles. Similarly, public vehicles, including, but not limited to, police cars, ambulances, and fire brigades have higher trust values in contrast to private vehicles. Moreover, drivers with many years of driving experience generally have higher trust values than novice drivers. This research, therefore, regards two contextual factors, i.e., vehicle types and operating scenarios, in order to obtain a more optimal trust value. Therefore, not only were the direct trust and the indirect trust included in the global trust of our envisaged trust model but the context was also incorporated (see Figure 2).

The salient contributions of the research-at-hand are as follows:We propose a novel trust management mechanism that takes into account direct trust (encompassing the trust attributes of interaction success rate, similarity, familiarity, and reward and punishment), indirect trust (involving recommendations via the one-hop neighboring nodes of a trustor pertinent to a trustee and the confidence of a trustor on the recommendations ascertained by the one-hop neighboring vehicles), and context (comprising vehicle types and operating scenarios) in order to ascertain the trust of vehicles in an IoV network.In contrast to the conventional trust management heuristics, we envisage a machine learning-based trust aggregation scheme in a bid to ascertain the optimal trust score of each vehicle in an IoV network so that they can be classified as either being trustworthy or untrustworthy.We carried out a comprehensive evaluation of our envisaged trust management mechanism and demonstrated that it outperforms other state-of-the-art trust management mechanisms.

The rest of the paper is systematically organized as follows. Section 2 delineates the state-of-the-art of trust management in IoV networks. Section 3 presents our envisaged trust management mechanism. Section 4 presents the experimental results and discussions pertinent to the same, and Section 5 concludes the paper.

## 2. Related Works

In recent years, there has been an increase in research on trust management in the IoV [2,5,11,19,20,21,22,23,24]. We can divide the existing trust management models into: learning-based and traditional methods-based trust management models.

### 2.1. Trust Parameters and Evaluation Parameters

A holistic overview of the existing trust management mechanisms reveals that a number of trust-based parameters have been applied in different settings in order to measure and evaluate trust. The trust-based parameters include, but are not limited to, resource availability [6], similarity [19,25,26], familiarity [6,7,22,27], timeliness [7], context [19,20,28,29], cooperativeness [19,30], community-of-interest (CoI) [19,30], confidence [31,32], reward [28,33], attitude, subjective norms, and perceptual behavioral control [5], freshness of data [34], and packet delivery ratio [7,25,31,35,36]. Also, the selection of a trust-based threshold for determining trustworthy and untrustworthy behavior is crucial. If the threshold is set too high by the system designers, the trustworthy nodes may be even removed from a network. Alternatively, if the threshold is set too low, the untrustworthy nodes would slowly jeopardize the entire network. A comparative summary of the trust parameters employed in the representative literature is depicted in Table 1.

It is interesting to note that context is the most frequently used trust parameter followed by cooperativeness, similarity and reward. Context is an extremely important trust parameter and a number of other trust parameters depend on the same, and are, therefore, dissimilar in different contexts. For instance, the number of interactions between vehicles is different in urban and highway scenarios. Also, different types of vehicles, i.e., high-priority vehicles, public transport vehicles, professional vehicles, and novice vehicles, have different trust values. Accordingly, this paper proposes a context-based trust management model. Table 1 further depicts that there are not many trust parameters employed in the state-of-the-art trust management models. If there are few trust parameters, the accuracy of the global trust value calculated will also decrease. To mitigate this problem, we propose a trust management model that includes six trust parameters, i.e., interaction success rate, similarity, familiarity, reward and punishment, confidence, and context.

There are two important steps in trust management, i.e., building a trust model and evaluating a trust model [31]. The purpose of trust evaluation is to evaluate the accuracy, reliability, and practicality of an envisaged trust model. The literature suggests that typical evaluation parameters employed for the trust-based models include precision, recall, and F1-score [19,20,29,37], false positive rate, true positive rate, true negative rate [27], and computation overhead [38].

**Table 1 sensors-24-00863-t001:** Trust parameters in trust management model.

References	Similarity	Familiarity	Timeliness	Context	Cooperativeness	CoI	Confidence	Reward
[2]	-	-	-	✓	-	-	-	-
[7]	-	✓	-	✓	✓	-	✓	-
[19]	✓	-	-	-	✓	✓	-	-
[20]	-	-	-	-	✓	✓	-	✓
[23]	-	-	-	✓	-	-	-	-
[24]	-	-	✓	✓	-	-	-	-
[25]	✓	✓	-	✓	-	-	-	-
[26]	✓	-	-	-	-	-	-	
[27]	✓	✓	-	✓	-	-	-	-
[33]	-	-	-	-	-	-	-	✓
[28]	-	-	-	-	-	-	-	✓
[29]	-	-	-	✓	✓	-	-	-
[31]	-	-	✓	✓	-	-	-	-
[32]	-	-	-	✓	✓	-	✓	-
[34]	-	✓	-	-	✓	-	-	-
[39]	-	-	-	✓	-	-	-	✓
[40]	✓	-	-	-	-	-	-	-
[41]	-	-	-	-	-	-	✓	-
[42]	-	-	-	-	-	-	-	✓
Our scheme	✓	✓	-	✓	✓	✓	✓	✓

### 2.2. Conventional Trust Management Models

A trust evaluation algorithm has been proposed in [5] that exploited the attributes (attitude towards behavior, subjective norms, and perceived behavioral control) from the theory of planned behavior, i.e., a human psychological theory, to ascertain the trustworthiness of vehicles in a vehicular network within a given context and to decide whether to accept or not the traffic-related warning messages from a particular vehicle. Moreover, the notion of fuzzy logic has been employed in a bid to segregate the vehicles’ trust levels as CompleteTrust, MediumTrust, and DisTrust. The effectiveness of the trust evaluation algorithm was verified via false positive rates, true positive rates, and F1-score vis-à-vis different proportions of malicious vehicles.

A context-aware and attack-resistant trust model for the IoV networks has been suggested in [16]. This model takes into account (a) local trust encompassing the weighted sum of both direct trust (packet delivery ratio and time decay) and indirect trust (confidence factor) and (b) context-dependent trust (propagation delay, cooperativeness, and familiarity). Also, the notion of an adaptive misbehavior detection threshold has been proposed to segregate malicious vehicles from dishonest vehicles. Moreover, the resilience against on–off attacks and the selective node attacks has been demonstrated by employing optimal and rational influencing parameters as weights during the process of the weights’ assignment.

A forest fire model has been proposed in [23] to select the minimum number of competent nodes suitable for broadcasting emergency messages in an IoV network. At first, a social community is established by calculating the similarity of the social characteristics between the nodes. Subsequently, some key factors, including, but not limited to, the number of connections, velocity of nodes, general activity of the nodes, and data forwarding capability of neighboring nodes, are used to select the core node and the complementary node for the dissemination of emergency messages within the established social community. This establishes a trust estimation and management mechanism for nodes based on their behavior in an IoV network. Experimental results suggest that this particular model demonstrated high accuracy under a high density of malicious nodes.

A novel hybrid trust management scheme for an IoV network has been proposed in [43] to evaluate both node-centric and data-centric trust. Node-centric trust has been determined by employing the distance between the message sender and the message evaluator in tandem with the antenna height of the message sender and the message evaluator, whereas data-centric trust has been ascertained by means of information quality and effective distance (via a tier-based approach) between the message sender and the message evaluator. A trust threshold has been further employed which facilitates rewarding (incrementing) and penalizing (decrementing) the trust score of the message sender. The performance of the trust management scheme has been evaluated under man-in-the-middle attacks and zigzag attacks.

### 2.3. Machine Learning-Based Trust Management Models

A trust computational heuristic model has been envisaged in [19] to establish trustworthy relationships among the physical objects, i.e., devices, and for mitigating potential risks throughout the decision-making process in an SIoT environment. The direct trust of a particular object (trustee) is ascertained by taking into consideration the trust attributes of friendship similarity, community of interest, cooperativeness, and reward/punishment. The indirect trust, on the other hand, is computed by requesting the direct trust from the nodes that have interacted with the trustee. The authors exploited the notion of machine learning to (a) aggregate the trust attributes in order to determine an optimal trust score and (b) determine the best possible boundary to segregate between the trustworthy, untrustworthy, and neutral interactions. The neutral interactions are later classified as trustworthy or untrustworthy via a percentage threshold mechanism so these interactions can be employed for real-world applications.

A quantifiable trust assessment model based on machine learning has been proposed in [20] to make decisions autonomously, i.e., without human intervention, in an IoT network. This model encompasses trust features, i.e., co-location relationship, co-work relationship, cooperativeness-frequency-duration, reward system, mutuality and centrality, and community of interest, in order to assess the knowledge of a trustor towards a trustee. These trust parameters are aggregated via machine learning to obtain a single trust value for each pair of nodes (trustor and trustee) and which are then further segregated into trustworthy and untrustworthy interactions via a decision boundary.

A trustworthy object classification framework, Trust-SIoT, has been further proposed in [29] to establish and maintain a trustworthy relationship between the IoT objects over time. The authors employed social characteristics of objects in the form of direct trust metrics, reliability and benevolence, credible recommendations, and the degree of relationships. A SIoT knowledge graph was further constructed in order to record five dynamic social relationships, including, but not limited to, co-location object relationships, parental object relationships, ownership object relationships, social object relationships (SOR), and a variant of SOR (to connect public and private mobile devices) to ascertain the degree of relationships. An artificial neural network-based model was further employed for decision-making purposes, i.e., to identify the trustworthiness level of a trustee. The performance of this framework is evaluated in terms of F1-score, MAE, and MSE.

Similarly, a machine learning-based trust model (encompassing trust parameters of similarity, familiarity, and packet delivery ratio) has been put forward in [25] to identify and eliminate malicious vehicles within an IoV network. A context-aware trust management framework for a VANET network has been suggested in [39] to ascertain the trustworthiness of messages received by vehicles to guarantee that no false information influences any driving decision-making process. This framework was composed of three modules, namely, information formalization, trust evaluation and strategy adjustment. The authors proposed a trust evaluation method based on evaluation strategy in different scenarios. In addition, information entropy theory was introduced into the trust calculation function to ensure more accurate evaluation results. Finally, a reinforcement learning model was proposed, and the evaluation strategy was dynamically adjusted according to the feedback of previous evaluation results.

To sum up, over the years, a number of both conventional and machine learning-based trust management algorithms have been proposed in the research literature, and which have laid the foundations for the research envisaged in this paper. Whilst these research papers have made some outstanding contributions, they still lack the potential of being a generic algorithm that can be suitably adapted to a particular research domain. In addition, they only take into consideration the traditional and limited trust parameters and a number of them even do not consider the influence of context on the trust values. Therefore, this research paper envisages a machine learning-based trust management mechanism that considers key influential trust parameters and the context for ascertaining the trust of vehicles in an IoV network.

## 3. Proposed Trust Evaluation Model

We hereby design a novel trust management framework, as depicted in Figure 3, in a bid to ascertain the trustworthiness of vehicles in an IoV network. The envisaged trust model primarily encompasses the following three salient steps:


*Step 1—Establishing the Trust Model*
The trust of any particular vehicle (trustee) is ascertained via a trust model which takes into account direct trust, indirect trust, and context. The direct trust is a trustor’s direct observation pertinent to a trustee and is composed up of four parameters, i.e., interaction success rate, similarity, familiarity, and reward and punishment. On the contrary, the indirect trust is computed via the respective trustor’s one-hop neighbors’ recommendations pertinent to a trustee and the degree of confidence of the respective trustor on the recommendations of its corresponding one-hop neighbors. It is also pertinent to mention that the model further takes into consideration the impact of context (vehicle types and operating scenarios) of both trustor and trustee.
*Step 2—Training the Trust Model*
Once the trust values have been computed via the trust model, we first employ unsupervised learning algorithms such as k-means, fuzzy c-means, and agglomerative (hierarchical) clustering, in order to ascertain two clusters, i.e., trustworthy and untrustworthy. Simply put, an unsupervised learning algorithm has been employed here to label the feature matrices ascertained in Step 1. Subsequently, we use supervised learning algorithms such as the k-nearest neighbors algorithm and random forest algorithm for training with 5-fold cross-validation so as to identify the optimal trust boundary for distinguishing between trusted and untrusted vehicles.
*Step 3—Evaluating the Trust Model*
The evaluation parameters, i.e., precision, recall, and F1-score are used for evaluating the performance of the envisaged IoV-based trust model.

We, therefore, define a set of vehicles $V_{m}$, $m=\left\{1,2,...,M\right\}$, comprising both trustworthy (honest) as well as untrustworthy (malicious) vehicles. At every time instance $t^{\prime }$, $t^{\prime }=\left\{1,2,...,t\right\}$, each vehicle interacts with vehicles in its immediate area to evaluate their trust based on the underlying interaction. This interaction takes place between a pair of a trustor *i* and a trustee *j*. The definitions of trust parameters employed in this section are delineated in Table 2.

### 3.1. Direct Trust ($T_{d(i,j,t)}$)

Direct trust refers to a trustor’s direct observation of a trustee. However, it is pertinent to mention that the historical interactions between a trustor and a trustee should also be taken into consideration, i.e., in addition to the current interaction, for ascertaining the trust of a trustee since a malicious vehicle may behave intelligently by altering between a malicious and a non-malicious behavior. In our envisaged model, we employ four key trust parameters, i.e., *interaction success rate*, *similarity*, *familiarity*, *reward and punishment*, in order to ascertain the direct trust between a trustor *i* and a trustor *j*. The details of these parameters are as follows:*Interaction Success Rate (ISR)*—The $ISR_{i,j,t}$ ($0\leq ISR_{i,j,t}\leq 1$) manifests the degree of interaction between a trustor *i* and a trustee *j* in an IoV network, and is depicted as:
(1)$$ISR_{i,j,t}=\frac{\sum_{t^{\prime }=1}^{t}R_{i,j,t^{\prime }}}{\sum_{t^{\prime }=1}^{t}S_{i,t^{\prime }}}$$
where $\sum_{t^{\prime }=1}^{t}R_{i,j,t^{\prime }}$ signifies total number of messages successfully received by a trustee *j* from a trustor *i* and $\sum_{t^{\prime }=1}^{t}S_{i,t^{\prime }}$ represents the total number of messages sent by the trustor *i* over the said time period.*Similarity (Sim)*—The similarity ($0\leq Sim_{i,j,t}\leq 1$) itself is a weighted amalgamation of external similarity (ES) and internal similarity (IS). The external similarity herein implies the degree of similar content accessed by a trustor *i* and a trustee *j* over the time *t* (Equation (Equation 3)), whereas the internal similarity represents the exchange of information, i.e., position, direction, and velocity between a trustor *i* and a trustor *j* (Equation (Equation 4)).
(2)$$Sim_{i,j,t}=w_{ES}ES_{i,j,t}+w_{IS}IS_{i,j,t}$$
where $w_{ES}$ and $w_{IS}$ refers to the weight of the $ES_{i,j,t}$ and $IS_{i,j,t}$, respectively, ($w_{ES}+w_{IS}=1$). The $ES_{i,j,t}$ and $IS_{i,j,t}$ are ascertained as:
(3)$$ES_{i,j,t}=\sum_{t^{\prime }=1}^{t}w_{E_{s_{t^{\prime }}}}ES_{i,j,t^{\prime }}$$
(4)$$IS_{i,j,t}=\sum_{t^{\prime }=1}^{t}w_{I_{s_{t^{\prime }}}}IS_{i,j,t^{\prime }}$$
where $w_{E_{s_{t^{\prime }}}}$ and $w_{I_{s_{t^{\prime }}}}$ manifests the weights of $ES_{i,j,t^{\prime }}$ and $IS_{i,j,t^{\prime }}$, respectively, at a time t′ ($w_{E_{s_{t^{\prime }}}}+w_{I_{s_{t^{\prime }}}}=1$). The $ES_{i,j,t^{\prime }}$ and $IS_{i,j,t^{\prime }}$ are ascertained as:
(5)$$ES_{i,j,t^{\prime }}=\left\{\begin{array}{cc} 1, & \mathrm{if}\;C_{v_{i,t^{\prime }}}=C_{v_{j,t^{\prime }}}\\ 0, & \mathrm{if}\;C_{v_{i,t^{\prime }}}\neq C_{v_{j},t^{\prime }} \end{array}\right.$$
where $C_{v_{i,t^{\prime }}}$ and $C_{v_{j,t^{\prime }}}$ implies the content accessed by a trustor *i* and a trustee *j*, respectively. Similarly, the $IS_{i,j,t^{\prime }}$ is computed as:
(6)$$IS_{i,j,t^{\prime }}=\frac{Pos_{i,j,t^{\prime }}+Dir_{i,j,t^{\prime }}+Vel_{i,j,t^{\prime }}}{3}$$
(7)$$Pos_{i,j,t^{\prime }}=\left\{\begin{array}{cc} 1, & \mathrm{if}\;Pos_{i,t^{\prime }}=Pos_{j,t^{\prime }}\\ 0, & \mathrm{if}\;Pos_{i,t^{\prime }}\neq Pos_{j,t^{\prime }} \end{array}\right.$$
(8)$$Dir_{i,j,t^{\prime }}=\left\{\begin{array}{cc} 1, & \mathrm{if}\;Dir_{i,t^{\prime }}=Dir_{j,t^{\prime }}\\ 0, & \mathrm{if}\;Dir_{i,t^{\prime }}\neq Dir_{j,t^{\prime }} \end{array}\right.$$
(9)$$Vel_{i,j,t^{\prime }}=\left\{\begin{array}{cc} 1, & \mathrm{if}\;Vel_{i,t^{\prime }}=Vel_{j,t^{\prime }}\\ 0, & \mathrm{if}\;Vel_{i,t^{\prime }}\neq Vel_{j,t^{\prime }} \end{array}\right.$$
where $Pos_{i,t^{\prime }}$, $Pos_{j,t^{\prime }}$, $Dir_{i,t^{\prime }}$, $Dir_{j,t^{\prime }}$, $Vel_{i,t^{\prime }}$, and $Vel_{j,t^{\prime }}$ represent the position, direction, and velocity, respectively, of a trustor *i* and a trustee *j* at a time $t^{\prime }$.*Familiarity (Fam)*—The familiarity ($0\leq Fam_{i,j,t}\leq 1$) is also segregated into external familiarity (EF) and internal familiarity (IF). The external familiarity refers to the ratio of the number of common vehicles interacting with a trustor *i* and a trustee *j* to the total number of vehicles interacting with a trustor *i* over the time *t*, i.e., the more the number of common interacting vehicles, the higher the familiarity between a trustor *i* and a trustee *j*. On the contrary, the internal familiarity signifies the interaction frequency between a trustor *i* and a trustor *j* over the time *t*, i.e., the higher the interaction frequency, the higher is the familiarity between the two. The same is illustrated in Equations (Equation 10)–(Equation 12).
(10)$$Fam_{i,j,t}=w_{EF}EF_{i,j,t}+w_{IF}IF_{i,j,t}$$
where $w_{EF}$ and $w_{IF}$ refers to the weight of $EF_{i,j,t}$ and $IF_{i,j,t}$, respectively, ($w_{EF}+w_{IF}=1$). The $EF_{i,j,t}$ is ascertained as:
(11)$$EF_{i,j,t}=\frac{\sum_{t^{\prime }=1}^{t}F_{i,j,t^{\prime }}}{\sum_{t^{\prime }=1}^{t}F_{i,t^{\prime }}}$$
where $\sum_{t^{\prime }=1}^{t}F_{i,j,t^{\prime }}$ represents the number of common interacting vehicles of a trustor *i* and a trustee *j*, whereas $\sum_{t^{\prime }=1}^{t}F_{i,t^{\prime }}$ is the total number of vehicles interacting with *i*. Similarly, the $IF_{i,j,t}$ is computed as:
(12)$$IF_{i,j,t}=\frac{\sum_{t^{\prime }=1}^{t}I_{i,j,t^{\prime }}}{t}$$
where $\sum_{t^{\prime }=1}^{t}I_{i,j,t^{\prime }}$ signifies the number of interactions between a trustor *i* and a trustee *j*.*Reward and Punishment (RP)*—The RP is employed to evaluate the rewards and punishments accorded to a trustee *j* by a trustor *i* depending on its behavior, i.e., a trustee *j* is rewarded by a trustor *i* for its cooperation, honesty, and reporting a critical event, and is punished for any misconduct. The RP is, therefore, calculated as:
(13)$$RP_{i,j,t}=ISR_{i,j,t}e^{-\frac{N_{p}}{N_{p}+N_{r}}}$$
where $ISR_{i,j,t}$ is the interaction success rate between a trustor *i* and a trustor *j*. Also, $N_{p}$ suggests the number of negative interactions, whereas $N_{r}$ exhibits the number of positive interactions.

### 3.2. Indirect Trust ($T_{ind(i,j,t)}$)

The indirect trust, also generally referred to as the recommendation trust, is ascertained by (a) soliciting the recommendations via the one-hop neighboring nodes of a trustor pertinent to a trustee and (b) by taking into account the confidence of a trustor on the recommendations ascertained by the one-hop neighboring vehicles [32,44]. The indirect trust is computed as:(14)$$T_{ind(i,j,t)}=\frac{\sum_{k=1}^{n}C_{i,k,t}T_{d\left(k,j,t\right)}}{n}$$
where $C_{i,k,t}$ implies the confidence score assigned by a trustor to the recommendations of its one-hop neighboring vehicles pertinent to a trustee, $T_{d\left(k,j,t\right)}$ refers to the recommendations ascertained by the said one-hop neighboring nodes, and *n* implies the total number of one-hop neighboring nodes. The confidence score, $C_{i,k,t}$, is calculated as:(15)$$C_{i,k,t}=\left\{\begin{array}{cc} 1, & \mathrm{if}\;T_{d(i,k,t)}\geq Th_{C}\\ 0.5, & \mathrm{if}\;Th_{T}\leq T_{d(i,k,t)}<Th_{C}\\ 0, & \mathrm{if}\;T_{d(i,k,t)}<Th_{T} \end{array}\right.$$
where $Th_{C}$ and $Th_{T}$ refer to the confidence threshold and the trust threshold, respectively, and act as a weight for distinguishing between a good, an average, or a bad recommendation [39].

### 3.3. Context ($T_{c}$)

A number of existing trust models ignore the significance of context, thereby making them quite unrealistic for real-world settings. In our model, the notion of context has been primarily determined by two factors, i.e., the vehicle types and the operating scenarios, the details of which are as follows:*Vehicle types (VT)*—Five types of vehicles have been taken into consideration in the proposed model. Police cars, ambulances, and fire engines are regarded as high-priority (HP) vehicles since the information disseminated by these particular vehicles possesses considerable confidence of a centralized trusted authority. The second type is public transport (PT) vehicles, i.e., buses, taxis, and subways, which are also considered reasonably trustworthy since they have been approved by specific authorized departments. Similarly, private vehicles are classified into professional (P) vehicles and novice (N) vehicles primarily depending on their respective driver’s driving experience, i.e., professional drivers are regarded to have extensive driving expertise in contrast to beginners and are, therefore, considered to be more trustworthy. Finally, we consider malicious vehicles to be untrustworthy in nature. Equation (Equation 16) illustrates the trust values vis-à-vis the suggested vehicle types:
(16)$$T_{VT}=\left\{\begin{array}{cc} 1, & \mathrm{if}\;\mathrm{Vehicles}\;=\;\mathrm{HP}\\ 0.8, & \mathrm{if}\;\mathrm{Vehicles}\;=\;\mathrm{PT}\\ 0.6, & \mathrm{if}\;\mathrm{Vehicles}\;=\;P\\ 0.4, & \mathrm{if}\;\mathrm{Vehicles}\;=\;N\\ 0, & \mathrm{if}\;\mathrm{Vehicles}\;=\;\mathrm{Malicious} \end{array}\right.$$

*Operating Scenarios (OS)*—In the envisaged model, we have considered two operating scenarios, i.e., an urban and a highway one. In Section 4, the simulation results for these two scenarios have been delineated in detail. It is pertinent to highlight here that the high mobility and the random geographical distribution of vehicles in an IoV network results in several different contextual scenarios. Therefore, it is indispensable to consider such settings while ascertaining the trust of a trustee. For instance, owing to the limited mobility of vehicles and the high density of RSUs in an urban scenario, there is a considerable number of interactions, both trustworthy and untrustworthy, between the vehicles. However, in scenarios involving highways, the mobility of the vehicles is generally much higher than that in an urban scenario. Furthermore, vehicles in highway settings have a more sparse geographical distribution, thereby resulting in fewer interactions between them. Trust management often relies on a large number of RSUs, but there are fewer RSUs on highways, so trust management cannot be well implemented in this scenario.

## 4. Results and Discussion

### 4.1. Simulation Setup and Feature Extraction

We used the Epinions dataset (https://cse.msu.edu/tangjili/datasetcode/truststudy.htm, Accessed: 1 June 2023) in order to map the data traces for the trust parameters of our envisaged IoV-based trust model. Epinions, in essence, is a publicly available trust dataset that encompasses six parameters: userid, productid, categoryid, rating, helpfulness, and timestamps. For instance, a data trace of [1, 2, 3, 4, 5, 6] in the Epinions dataset implies that user 1 accords a rating of 4 to product 2 belonging to category 3 at timestamp 6. The helpfulness of the accorded rating is 5. For the sake of the research at hand, we have appropriately transformed the Epinions dataset into an IoV dataset in light of the similar transformations envisaged in [45].

A total of 3266 pairs of interactions between trustors and trustees, i.e., pertinent to 64 nodes (vehicles), have been taken into consideration. The same are arranged in the form of a feature matrix *M* as illustrated in Equation (Equation 17).
(17)$$\left[M_{n\times 7}\right]={\left[\begin{array}{ccccccc} ISR_{1} & Sim_{1} & Fam_{1} & RP_{1} & confidence_{1}\; & VT_{1} & OS_{1}\\ \vdots & \vdots & \vdots & \vdots & \vdots & \vdots & \vdots \\ ISR_{n} & Sim_{n} & Fam_{n} & RP_{n} & confidence_{n} & VT_{n} & OS_{n} \end{array}\right]}_{n\times 7}$$

The dimension of this feature matrix is n×7, wherein *n* = 3266 and 7 implies the trust-based feature vectors via-à-vis each of the 3266 trustor trustee pairs. It is pertinent to mention here that there is no need for the features’ normalization since each trust feature value falls in the range of [0, 1]. The seven features are concatenated into three features, i.e., direct trust, indirect trust, and context, in a bid to form a new feature matrix *N* with a dimension of n×3, wherein the direct trust implies interaction success rate, similarity, familiarity, and reward and punishment, the indirect trust is ascertained via direct trust and confidence, and the context comprises vehicle types and operating scenarios. Since it is not feasible to display a three-dimensional vector, two out of three features are selected and displayed at a time for demonstration purposes.
(18)$$\left[N_{n\times 3}\right]={\left[\begin{array}{ccc} direct\;trust_{1} & indirect\;trust_{1} & context_{1}\\ \vdots & \vdots & \vdots \\ direct\;trust_{n} & indirect\;trust_{n} & context_{n} \end{array}\right]}_{n\times 3}$$

Table 3 depicts the trust-based parametric values of 20 randomly selected vehicles in the IoV network. Figure 4 further portrays two of such parameters, i.e., ISR and RP, for all of the 64 vehicles in the IoV network. It is evident that the change in the parametric values of RP is proportional to the parametric values of ISR with the exception of a few. For instance, vehicles 3, 32, 48, 52, and 58 possess high ISR values but low RP values. This is owing to the fact that although the interactions carried out by these vehicles are considerable, most of them were accounted for as being negative.

### 4.2. Clustering and Labeling

Subsequent to the extraction of the desired trust features, three unsupervised learning algorithms, i.e., k-means, fuzzy c-means, and agglomerative clustering, have been employed to label the feature matrices. It is noteworthy that unsupervised learning algorithms have been employed in a bid to ascertain a credible and reliable ground truth. Accordingly, two clusters, trustworthy and untrustworthy, have been obtained as a result of the same and are depicted in Figure 5, Figure 6 and Figure 7.

### 4.3. Classification and Model Evaluation

A number of supervised learning algorithms, including, but not limited to, k-nearest neighbor (KNN), support vector machine (SVM), random forest (RF), and ensemble ones have been employed in the research literature for classification purposes [7,19,46,47]. For the manuscript at hand, we have employed KNN and RF classifiers on the resulting feature matrix for training purposes via a 5-fold cross-validation approach in a bid to ascertain the malicious nodes via a decision boundary. The same is depicted in Figure 8 and Figure 9 for KNN and RF classifiers, respectively, wherein the trusted and untrusted regions can be clearly observed. We have subsequently evaluated the accuracy of our envisaged trust model via the following three evaluation parameters:Precision: Precision depicts the ability of the envisaged trust model to correctly predict malicious vehicles as being malicious.Recall: Recall refers to the proportion of malicious vehicles that have been correctly ascertained by the envisaged trust model.F1-score: F1-score implies the weighted harmonic mean of the precision, and recall and ascertains the model’s accuracy.

For our trust model, we further consider vehicles under two different operating scenarios, i.e., urban and highway. Also, owing to the space constraint, only the figures pertinent to the urban scenario have been portrayed. Nevertheless, the precision, recall, and F1-score for both urban and highway scenarios have been depicted in Table 4. It is pertinent to mention here that the precision, recall, and F1-score of our envisaged trust model as demonstrated by the KNN classifier under both urban and highway settings is much higher in contrast to the precision, recall, and F1-score demonstrated by the RF classifier under the same settings. KNN is regarded as one of the simplest classification algorithms, i.e., with mature theory, low training time complexity, and insensitivity to outliers. This particular algorithm is quite suitable for an automatic classification of class domains with large sample size [7]. It is also noteworthy to mention that the trust can be ascertained in a relatively more accurate manner in an urban setting in contrast to the highway setting since vehicles interact much more frequently in the former owing to their low speeds as opposed to the latter which is designed to enable them to traverse with high speeds.

Table 5 depicts the comparison of our envisaged trust model vis-à-vis machine learning-based trust mechanisms, i.e., [47,48]—labeled as NC—1 and NC—2, respectively, that have not taken the notion of context into consideration. Whilst the said trust models demonstrate high precision, our envisaged trust model still outperforms them since it takes into account the context pertinent to the interactions on the premise that the interaction between a trustor and trustee is different in different contexts. Table 5 further outlines the comparison of our envisaged trust model vis-à-vis conventional (weighted sum) trust mechanisms, i.e., [23,27,49]—labeled as Conv1, Conv2, and Conv3, respectively. It can once again be seen clearly that the envisaged trust model performs considerably better in terms of precision in contrast to the conventional trust mechanisms. This reinforces the fact that the weighted sum mechanisms have strong subjectivity and are influenced by numerous underlying factors. Hence, when a number of trust parameters are in play, a machine learning-based mechanism is optimal for not only aggregating the same but ascertaining an intelligent trust boundary.

## 5. Conclusions and Future Directions

An intelligent transportation system is an intrinsic component of smart cities since it allows vehicles to employ vehicle-to-everything communication in a bid to exchange safety-critical messages with the other road entities and the supporting infrastructure to ensure highly secure and intelligent traffic flows. However, road entities within an IoV network are vulnerable to a number of attacks and malicious actors prevailing in the same are always on the lookout to manipulate the IoV network for their malicious gains. In this manuscript, a machine learning-based trust management mechanism, MESMERIC, has been proposed that takes into account direct trust, indirect trust, and context (each with a number of qualifying attributes) to not only ascertain the trust of vehicles in an IoV network but to segregate the trustworthy vehicles from the untrustworthy ones by means of an optimal decision boundary. In the near future, the authors would investigate designing and launching a number of dynamic trust-related attacks via a state-of-the-art trust-based IoV testbed in order to understand the underlying nitty gritty of such dynamic attacks so that more resilient IoV-based trust models could be formulated. Additionally, the authors aim to propose an intelligent weighting-based conventional mechanism in a bid to mitigate any possible subjectivity that could arise during the trust aggregation process.

## Figures and Tables

**Figure 1 sensors-24-00863-f001:**
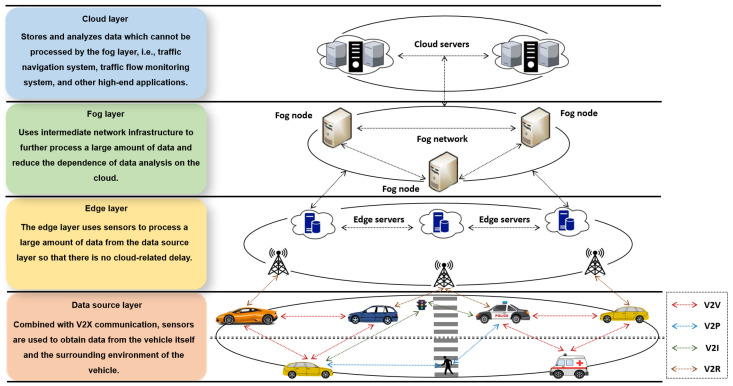
A system architecture of the IoV.

**Figure 2 sensors-24-00863-f002:**
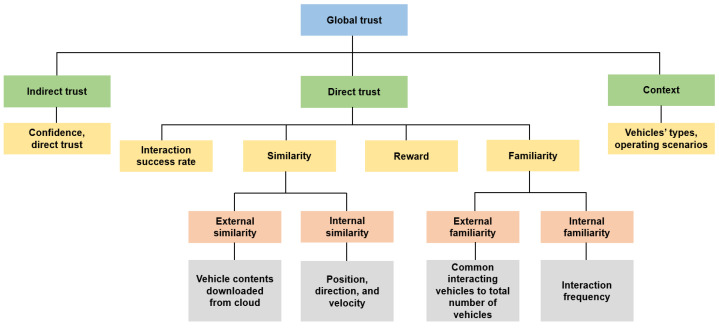
The composition of the global trust.

**Figure 3 sensors-24-00863-f003:**
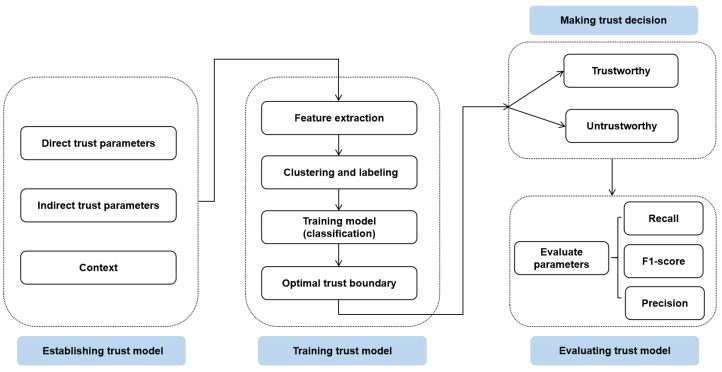
The framework of the proposed trust management model.

**Figure 4 sensors-24-00863-f004:**
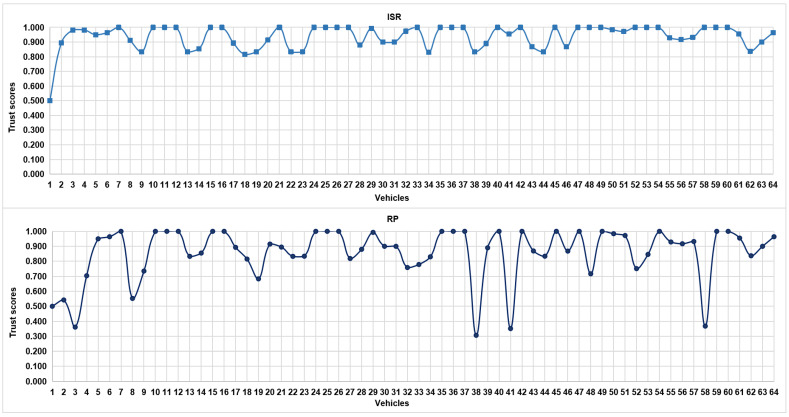
Trust scores of vehicles in an IoV network vis-à-vis ISR and RP (ISR here implies interaction success rate, and RP refers to reward and punishment).

**Figure 5 sensors-24-00863-f005:**
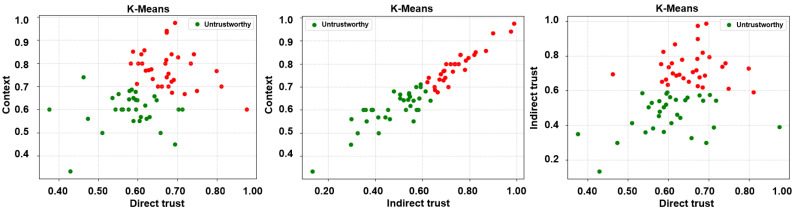
Labels via unsupervised learning (k-means clustering)—direct trust vs. context, indirect trust vs. context, and direct trust vs. indirect trust.

**Figure 6 sensors-24-00863-f006:**
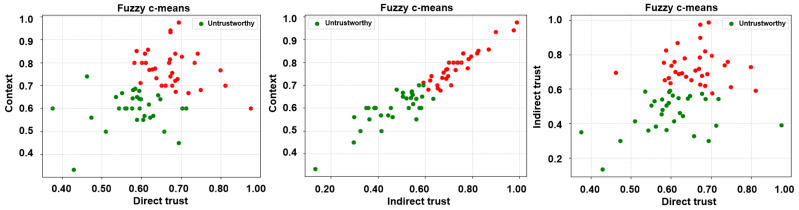
Labels via unsupervised learning (fuzzy c-means clustering)—direct trust vs. context, indirect trust vs. context, and direct trust vs. indirect trust.

**Figure 7 sensors-24-00863-f007:**
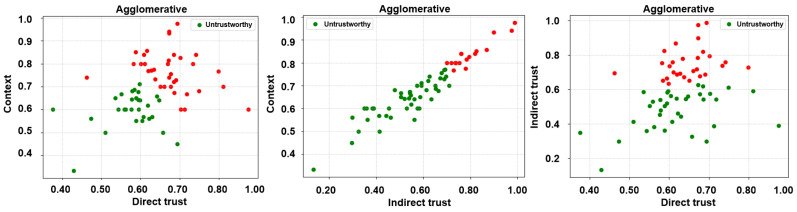
Labels via unsupervised learning (agglomerative clustering)—direct trust vs. context, indirect trust vs. context, and direct trust vs. indirect trust.

**Figure 8 sensors-24-00863-f008:**
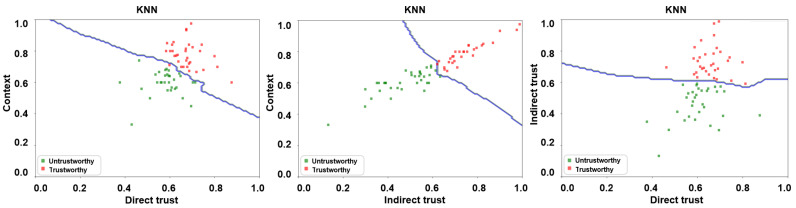
Trust boundary results for KNN algorithm—direct trust vs. context, indirect trust vs. context, and direct trust vs. indirect trust.

**Figure 9 sensors-24-00863-f009:**
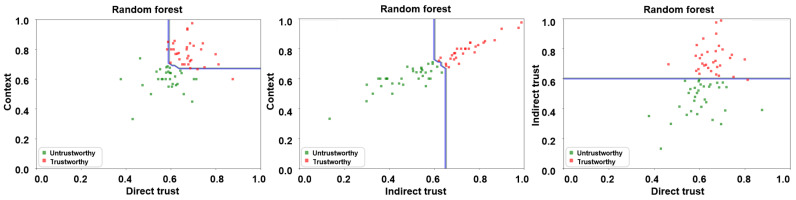
Trust boundary results for RF algorithm—direct trust vs. context, indirect trust vs. context, and direct trust vs. indirect trust.

**Table 2 sensors-24-00863-t002:** Mathematical symbols employed in the envisaged trust model.

Symbol	Definition
*i*	Trustor
*j*	Trustee
*k*	The neighbor of *i*
$t^{\prime }$	A time instance
*t*	The current time instance
$Th_{C}$	Confidence threshold (0.8)
$Th_{T}$	Trust value threshold (0.6)
ISR	Interaction success rate
Sim	Similarity
ES	External similarity
IS	Internal similarity
Fam	Familiarity
EF	External familiarity
IF	Internal familiarity
RP	Reward and punishment
VT	Vehicle types
OS	Operating scenarios
*n*	3266 pairs of interactions

**Table 3 sensors-24-00863-t003:** Trust parameters’ values pertinent to 20 random vehicles in an IoV network (ISR here implies interaction success rate, and RP refers to reward and punishment).

Vehicles	ISR	Similarity	Familiarity	RP	Confidence	Context
1	0.500	0.614	0.167	0.500	0.000	0.333
2	0.894	0.593	0.120	0.542	0.000	0.711
3	0.982	0.665	0.119	0.361	1.000	0.800
4	0.982	0.770	0.111	0.704	1.000	0.567
5	0.950	0.453	0.104	0.950	0.500	0.850
6	0.964	0.484	0.107	0.964	0.000	0.857
7	1.000	0.515	0.125	1.000	1.000	0.650
8	0.911	0.541	0.226	0.552	1.000	0.771
9	0.833	0.476	0.262	0.735	1.000	0.686
10	1.000	0.524	0.217	1.000	1.000	0.720
11	1.000	0.600	0.292	1.000	1.000	0.933
12	1.000	0.763	0.200	1.000	0.500	0.840
13	0.833	0.750	0.222	0.833	1.000	0.600
14	0.855	0.750	0.375	0.855	1.000	0.550
15	1.000	0.540	0.229	1.000	1.000	0.500
16	1.000	0.529	0.319	1.000	1.000	0.600
17	0.893	0.638	0.351	0.893	0.500	0.729
18	0.815	0.700	0.100	0.815	1.000	0.618
19	0.834	0.585	0.323	0.683	1.000	0.775
20	0.915	0.700	0.333	0.915	1.000	0.600

**Table 4 sensors-24-00863-t004:** Evaluation results via supervised learning algorithms, i.e., KNN and RF (KNN here implies k-nearest neighbor and RF refers to random forest).

Scenarios	Classifier	Precision	Recall	F1-Score
Urban	KNN	1.0000	1.0000	1.000
	RF	1.0000	0.9400	0.9684
Highway	KNN	0.9804	0.9623	0.9713
	RF	0.9764	0.9338	0.9546

**Table 5 sensors-24-00863-t005:** Comparison of the precision of trust models (NC—1: [47], NC—2: [48], Conv1: [23], Conv2: [27], Conv3: [49]).

Model	Proposed	NC—1	NC—2	Conv1	Conv2	Conv3
Precision	1.0000	0.9234	0.9005	0.9700	0.9700	0.9750

## Data Availability

The data presented in this study are available on request from the corresponding author.

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
