# Peer review of "MESMERIC: Machine Learning-Based Trust Management Mechanism for the Internet of Vehicles"

_sensors, 2024, doi:10.3390/s24030863_

Round 1
Reviewer 1 Report
Comments and Suggestions for Authors
1. Similar to traditional methods, the proposed mechanism may still face challenges related to subjective setting of weights for trust parameters. The authors should develop a more objective approach to set weights for trust parameters, reducing subjectivity and improving consistency.
2. If the weight parameter set unreasonable, formula (3) and (4) in the value of the cumulative after more than 1, then for the value of formula (2) will not be possible more than 1, if possible, with the conditions do not match.
3. The article only gives a description of the model, the algorithm for training should also be given a brief overview.
4. The following work related to Machine-learning based performance improvement for IoV is missed,
“Towards V2I Age-aware Fairness Access: A DQN Based Intelligent Vehicular Node Training and Test Method”, Chinese Journal of Electronics, vol. 32, no. 6, 2023, pp. 1230-1244.
Reviewer 2 Report
Comments and Suggestions for Authors
In the manuscript, the authors have proposed a machine learning-based trust management mechanism, MESMERIC, which jointly considers direct trust, indirect trust, and context. Simulations have been conducted based on the Epinions dataset to evaluate the proposed mechanism. The manuscript is well organized overall. However, some concerns still need to be addressed.
1. It is informal to use the citation [x] as the subject of a sentence.
2. In Table 2, t and t’ are defined as the time instance and time point, respectively. The definitions are incorrect according to the usage of these two notations in Eq. (1).
3. For the description of Familiarity, the external familiarity is defined based on the relationship between trustor i and trustee j. The internal familiarity is defined based on the relationship between trustor i and trustor j, instead of trustee j. Could the authors please provide further explanations on the usage of trustor j? It is not easy to follow.
4. In Line 349 and Line 353, where does not need to be indented.
5. From Fig.5-7, it can be seen that different clustering algorithm lead to slightly different labeling results. How to evaluate the labeling result? What is the impact of the labeling result on the final classification result?
6. In Table 5, the authors compared the trust models. Why did the authors merely compare the precision? How about the F1-score and recall?
Comments on the Quality of English LanguageModerate editing of English language required
Reviewer 3 Report
Comments and Suggestions for Authors
The comments are as follows,
1. There are some problems with the format:
a) The sizes of the figures and tables are inconsistent, some of them are too large and some are not centered.
2. The length of the Section 1 and 2 makes the manuscript look like overview/research, this part needs to be drastically cut.
3. Figure 1 and Figure 2 have too much text and are not intuitive enough. It is suggested to improve them.
4. The authors' experimental results are too perfect (e.g., 100% Precision) with using some complex machine learning tools, which indicate that there may exist overfitting, please discuss it.
Round 2
Reviewer 1 Report
Comments and Suggestions for Authors
This paper can be accepted.
Reviewer 2 Report
Comments and Suggestions for Authors
Many thanks for the efforts of the authors. Most of my concerns have been addressed. I have no further comments.
Comments on the Quality of English LanguageIt would be better to further polish the writing.
Reviewer 3 Report
Comments and Suggestions for Authors
No further comments.